# Nestedness-Based Measurement of Evolutionarily Stable Equilibrium of Global Production System

**DOI:** 10.3390/e23081077

**Published:** 2021-08-19

**Authors:** Jiaqi Ren, Lizhi Xing, Yu Han, Xianlei Dong

**Affiliations:** 1College of Economics & Management, Beijing University of Technology, Beijing 100124, China; renjq@emails.bjut.edu.cn (J.R.); hanyu@bjut.edu.cn (Y.H.); 2International Business School, Beijing Foreign Studies University, Beijing 100089, China; 3Business School, Shandong Normal University, Jinan 250358, China; sddongxl@sdnu.edu.cn

**Keywords:** global value chain, global economic integration, nestedness, evolutionarily stable equilibrium

## Abstract

A nested structure is a structural feature that is conducive to system stability formed by the coevolution of biological species in mutualistic ecosystems The coopetition relationship and value flow between industrial sectors in the global value chain are similar to the mutualistic ecosystem in nature. That is, the global economic system is always changing to form one dynamic equilibrium after another. In this paper, a nestedness-based analytical framework is used to define the generalist and specialist sectors for the purpose of analyzing the changes in the global supply pattern. We study why the global economic system can reach a stable equilibrium, what the role of different sectors play in the steady status, and how to enhance the stability of the global economic system. In detail, the domestic trade network, export trade network and import trade network of each country are extracted. Then, an econometric model is designed to analyze how the microstructure of the production system affects a country’s macroeconomic performance.

## 1. Introduction

Being relatively independent for a long period of time, the production systems of countries and regions in the world have gradually formed a global economic system through flourishing trade. Numerous industrial value chains (IVCs) scattered among countries or regions constitute the global value chain (GVC) network, which is an interconnected and organic whole with specific functions. In such a network, the industrial sectors of each country form an interdependent and competitive community of shared destiny through the flow of capital, material and information.

In theoretical ecology and evolutionary biology, “nestedness” refers to a structural measure of the overall stability in the ecosystems [1,2,3]. The structure is an optimal system state conducive to both sides, which is the result of a mutual benefit mechanism established between species and between species and the environment through evolutionary games [4,5]. Mutualistic species strengthen cooperation through network reciprocity and weaken competition by reducing niche overlap, so as to promote the system to an evolutionary equilibrium. This concept has, in recent years, also begun to be applied by sociologists and economists to analyze various phenomena related to human society. In the same way as the ecosystem, the economic system seeks an evolutionary equilibrium in the process of dynamic evolution. In today’s highly developed globalization, cooperation among economies has reached an unprecedented level. Through dynamic games to allocating scarce resources, economies in the global value chain can maximize their relative interests. The role of network reciprocity in emerging cooperation is an important mechanism for the global economic system to achieve dynamic balance [6]. As the leading link in the global value chain, the flow of intermediate products depends on the cooperation between various industrial sectors. Considering that the industrial sector in the GVC has a dual identity, the provider and consumer of intermediate products, it can be represented by a bipartite graph to separate the two attributes of a node, and, thus, the mutualistic relationship between upstream and downstream industries can be clearly depicted. In fact, the ecosystem and the global production system have some common ground. For instance, both the flow of energy between species and the flow of intermediate goods between industrial sectors reflect the mutually beneficial symbiosis relationship produced through the competition and cooperation game, as shown in Figure 1. With the nested structure being identified among industrial sectors in the GVC network, nestedness can be applied to measure the topological stability of both the whole and local parts of the global industrial ecosystem.

The ecological metaphor is not ecological reductionism or ecological imperialism, and it is not to simply reduce the phenomenon of macroeconomic evolution (industrial transfer between countries and adjustment of industrial structure within countries) to ecological evolution [7]. Moreover, Chase and Leibold’s research on the ecological niche is only an abstract milieu interne adjustment mechanism, and does not describe the specific evolution process. The complex system theory must be embedded in it to accurately explain the law of the evolution of an economic system [8]. Therefore, the evolutionary game theory of biological populations in ecology has a certain enlightening significance to the theory of economic evolution.

## 2. Literature Review

Nestedness, derived from theoretical ecology and evolutionary biology, is an important structural feature of complex networks. In 1957, Darlington mentioned this concept in his book *Zoogeography*, in which he noticed that the spatial distribution of species displayed the nested feature [9]. Building on Darlington’s discovery, in 1986, Patterson and Atmar formulated the precise conce pt of nestedness, i.e., in a fully nested network, the neighborhood of a node with lower node degree is a subset of the neighborhood of a node with higher node degree [10]. In 2003, Bascompte et al. analyzed 25 plant–pollinator networks and 27 plant–frugivore networks and found that most of the networks exhibited nested features [11]. The nested structure reflects the mutualistic relationship between species. In a mutualistic network, specialist species tend to relate to generalist species who have higher adaptability to the environment, thus mitigating the risk of extinction. Niche overlap decreasing in the nested structure helps weaken competition and improve species diversity [12], and the greater the nestedness, the stronger the recovery ability of the system after external shocks and the stabler the network structure [13,14,15].

Inspired by this discovery in ecological networks, scholars in socioeconomic networks began to devote themselves to the study of nestedness. As early as 1965, when studying the U.S. economic structure using intra-country input–output data, Leontief identified the obvious nestedness of the U.S. industrial network [16]. Subsequently, a large number of theoretical and empirical studies have emerged, which has greatly enriched the interdisciplinary research in the field of economics. The world trade network [17], the arms trading network [18], the interbank capital flow network [19], the manufacturer–supplier network [20] and the product export network [21,22,23] also show nested features. The network heterogeneity caused by the dynamic evolution of a social economic system is the main reason for its nested structure. Taking the world trade network as an example, the coexistence of competition and cooperation among countries leads to unbalanced economic development, which makes the world trade network show a center–periphery structure. That is, the generalist sectors are connected with the most counterparts and form the core of the network, and specialist sectors are at the periphery and are dependent on the center [24,25]. This highly connected center makes the links of the network replaceable. Even if the supply or demand of some sectors disappears, the existence of other replaceable sectors can make the products flow normally. At the same time, these studies reflect that the nested structure is of great significance for maintaining the stability of the economic system [26,27]. For example, in the 2008 global financial crisis, the reason for the decrease in interbank transactions was that the core banks reduced the number of externally active sides [28].

This paper is organized as follows. In Section 1, the application and development of nested structure theory in the field of ecology and economics are systematically introduced. In Section 2, we sort out the literature on nested structure in economic system and ecosystem. In Section 3, a GVC network is built based on the Multi-Region Input–Output (MRIO) database to embody the flow of intermediate goods between industrial sectors. In Section 4, the nested structure is embodied by sorting algorithm and measured by NODF method. In Section 5, analyses of the divergence, trend and stability are conducted to explain the complex relations between industrial sectors and the global production system, and then the economically evolutionary mechanism is proposed. In Section 6, the econometric models are used to analyze the relations between the nestedness-based indicators and the level of economic development. Finally, some countermeasures are put forward for economies to achieve a much more stable and healthy state.

## 3. Data and Model

In order to represent the nested structure of the global production system, we build a GVC network that can reflect the coopetition relationships of industrial sectors in each country.

### 3.1. Data Sources and Structure

Among the mainstream databases, the Eora Multi-Region Input–Output (MRIO) database of the University of Sydney covers the largest number of economies (189 countries/regions) and the longest period (1990–2015) and is suitable to be taken as modeling data. To make the nested structure more explicit and hierarchical, a simplified version of this database, named Eora26, is chosen in this paper, i.e., to sort the industrial sectors in 189 countries/regions into 26 sectors. For the sake of effectively visualizing the data element, we further group these sectors into four sectoral categories, namely agriculture (sectors 1 and 2), mining (sector 3), manufacturing (sectors 4 to 14), and services (sectors 15 and 16) (For detailed dividing criteria, refer to Appendix A).

### 3.2. Network Modeling

As an emerging yet important research area, GVC accounting is mainly represented by teams of Timmer M.P., Koopman R. and Wang Z., who have made important breakthroughs in economic theories and statistical techniques [29,30,31]. The important quantitative results they obtained have enriched the original GVC studies and cemented a theoretical basis for both the upcoming analysis and the formulation of relevant policies. They have also enabled the theoretical expansion to other GVC-related fields. Among all the achievements in GVC accounting, a set of preliminary accounting systems has been formed around value-added exports, which reflects industrial sectors’ competitiveness and participation in the GVC.

The global economic system, however, is a complex nonlinear emergence system, and the multiple emergences as its essential feature cannot simply be obtained by the linear addition of individualities. That is to say, the whole picture will be shadowed if only the individuals are analyzed. We should focus on the interrelationship and influence mechanism between individuals and the whole, under the perspective of systems science.

The fact that industrial sectors in the GVC function as both upstream and downstream sectors can be displayed in a bipartite graph, as in Figure 2d. Based on the data of intermediate use in the MRIO table, a Global Industrial Value Chain (Bipartite Graph) Network (GIVCNBG) is constructed in the form of a bipartite graph $G=\left(O,P,E,W\right)$. In the *G*, all upstream industrial sectors form the set of object nodes O, and all downstream industrial sectors form the set of participant nodes P; edges pointing from upstream to downstream form the set of edges E in competition with other N-1 sectors as a consumption sector, downstream industry sector *i* obtains from its upstream industry sector j intermediate goods input, the amount of which is$\;w_{ji}$ (j=
*i* indicates that the upstream and downstream are the same sector), constituting the weight set W.

While the GIVCNBG model applies the weight set W to substitute the adjacency matrix, each row refers to the distribution of intermediate goods output of an upstream sector and each column the intermediate goods input of a downstream sector. In spite of this being the same mechanism as the one-mode GIVCN model, a two-mode network is able to identify the underlying cooperative relationships between industrial sectors. That is, there is cooperation between upstream sectors to promote the production of their common downstream sectors [32]. The authors believe that the flow of intermediate goods in production systems (expressed in the IO table as value or currency flow) is similar to the flow of energy in ecosystems, and that both systems converge to a steady state after a complex game. As mentioned above, ecological studies have found that ecosystems in a steady state are characterized by nested structures and a more stable mutualistic relationship between species. Therefore, we believe such features can also be found in the topology of production systems due to them having the same evolutionary mechanism.

### 3.3. Network Pruning

The GIVCNBG model is an extremely dense weighted network with highly heterogeneous material flows between each upstream sector and downstream sector. Thus, it needs to be pruned in search of the backbone part before nested structure analysis. In this paper, a new heuristic algorithm—XIFA [33,34]—is proposed, with the integration of the features of H-index [35], Pareto Principle and Disparity Filter [36]. Through XIFA, a special subgraph $\overset{´}{G}=\left(O,P,\overset{´}{E},\overset{´}{W}\right)$ is extracted from the GIVCNBG model, named as the GIVCNBG-FE model (FE as in “Filtering Edges”).

The GIVCNBG-FE model compresses the size of the edge set E to a large extent, For example, after pruning the GIVCNBG-Eora26SC4-2015 model by XIFA, $\left|\overset{´}{E}\right|=8.95\%\times \left|E\right|$, while $\sum \overset{´}{w_{ij}}=99.15\%\times \sum w_{ij}$, which means more than 90% of the deleted edges carry less than 1% of the network information, leaving less than 10% of the edges carrying more than 99%. In sum, the bipartite graph $\overset{´}{G}=\left(O,P,\overset{´}{E}\right)$, with the weight information removed and all edges left as important, is sufficient to portray the nested structure of the network (For detailed algorithm information, refer to Appendix B).

## 4. Measurement

A nested structure is determined by the distribution of edges in the network and can be influenced by the network connectivity. The higher the connectivity of the network, the more likely it is to exhibit nested characteristics. In ecosystems, nested structure is established when ecological niches of different species adapt to each other and, thus, achieve dynamic equilibrium. It is a network structure characteristic formed by species adapting to the natural environment in pursuit of homeostasis. Nestedness in an ecological network is therefore a measure of the stability and sustainability of an ecological environment. From the perspective of bionomics, there are many similarities between a GVC network and an ecological network in terms of topological characteristics. In the same way as biological species, the industrial sectors in the GVC form a complex association of mutual benefit, and the trade and economic cycles between them make the GVC an organic whole. Higher nestedness of the GVC network indicates a more mature industrial trade mechanism, a more regular and orderly industrial trade network and a deeper integration between industries. Hence, research on the nested structure of the GVC network has fundamental implications for the economic development of countries, regions and even the world [37].

Prior to the analysis, the adjacency matrix needed to be reordered to maximize the degree of network nestedness. Several classical sorting algorithms are introduced in Appendix C, and the SBD (sorted by degree) algorithm was finally selected through comparing their NODF. The NODF metric proposed by Almeida-Neto et al. is widely used to calculate the nestedness of networks, which is based on two basic properties: Decreasing Fill (DF) and Paired Overlap (PO) [38].

Given that a matrix has m rows and n columns, and MT is the number of elements valued at 1 in any row or column. For any pair of rows $\left(i,\;j\right)(i<j)$, if $MT_{i}>MT_{j}$, then $DF_{ij}=100$, otherwise $DF_{ij}=0$; similarly, for any pair of columns $\left(k,\;l\right)(k<l)$, if $MT_{k}>MT_{l}$, then $DF_{kl}=100$, otherwise $DF_{kl}=0$.

For any pair of rows $\left(i,\;j\right)(i<j)$, $PO_{ij}$ refers to the percentage of 1′s in a given row j that is located at identical column positions to the 1′s observed in a row i. Similarly, for any pair of columns $\left(k,\;l\right)(k<l)$, $PO_{kl}$ refers to the percentage of 1′s in a given column l that is located at identical row positions to those in column k. Therefore, for any up-to-down row pair, or any left-to-right column pair, the degree of paired nestedness ($N_{paired}$) can be expressed as follows
(1)$$N_{paired}=\left\{\begin{array}{c} 0,\;if\;DF_{paired}=0\\ PO,\;if\;DF_{paired}=100 \end{array}\right.$$

There are $m\left(m-1\right)/2$ row pairs in row m, and $n\left(n-1\right)/2$ column pairs in column n. Thus, the nestedness of the entire network can be calculated by “averaging all paired values of rows and columns”
(2)$$NODF=\frac{\sum N_{paired}}{\left[\frac{m\left(m-1\right)}{2}\right]+\left[\frac{n\left(n-1\right)}{2}\right]}$$
where the *NODF* value ranges from 0 to 100, with NODF=0 indicating a non-nested network structure and NODF=100 indicating a fully nested network structure.

## 5. Results

Globalization is both an opportunity and a threat for the economic development of each country. On the one hand, the industrial sectors of each country have their comparative advantages, thus forming a relatively stable international industrial division of labor. On the other hand, they also fiercely compete in the global market, seeking a place on the GVC. It is under the impetus of both cooperation and competition that the global economic system evolves and shows nested structural characteristics in the process of convergence to homeostasis.

### 5.1. Divergence Analysis

If the economic system is compared to an ecosystem, the generalist feature of an industrial sector can be measured by the number of important IO relationships they establish with other sectors. In this paper, the larger-degree industrial sectors are defined as Generalist Sector, featuring higher involvement in the GVC, widely distributed outputs/inputs and a broader industrial ecological niche. On the opposite side are the Specialist Sector. Viewed by rows, the nodes in the upper part of the nested area have higher outdegree and stronger supply-side generalist degree, while those in the lower part have lower outdegree and weaker supply-side generalist degree. Viewed by columns, the nodes on the left side have higher indegree and stronger demand-side generalist degree, while those on the right side have lower indegree and weaker demand-side generalist degree.

Due to the vertical specialization, product manufacturing, and its related services, exists through all stages of the global production process. Each country takes advantage of its own and others’ comparative advantages in technology, capital and/or labor, jointly shaping the main structure of the GVC. As a result, the manufacturing and services sectors own a higher degree of external dependence, i.e., a higher generalist degree. In contrast, the agriculture and mining sectors only affect a limited number of sectors. They mainly trade with the domestic sectors for self-sufficiency and seldom establish international trade channels with the manufacturing sectors of a few developed economies. In short, most of them have low involvement in the GVC, resulting in lower generalist degree.

To further analyze the features of the nested structure of the GVC, this paper selected four representative areas consisting of the top twenty and bottom twenty sectors on the supply and demand sides, respectively, as shown in Figure 3. The comparison reveals significant differences in the generalist degree of industrial sectors in developed and developing countries.

Region A on the top left shows the IO relationship between the upstream and downstream generalist sectors. This value network, consisting of manufacturing and services sectors in advanced economies, is very dense, indicating that intense competition occurs because of overlapping ecological niches. The NODF value of Region A is 76.165, due to the empty elements of the upper triangle and the non-empty elements of the lower triangle, which indicate the insufficient collaboration and excessive competition among these generalist sectors; hence, negatively affecting the stability of the industrial structure.

Region B in the top right shows the IO relationship between the upstream generalist sectors and the downstream specialist sectors. It is found that Japan’s services belong to the generalist sector in the upstream, whose trade in services exports to most countries around the world, while they have very low generalist degree in the downstream. On the one hand, Japan’s services sectors are highly developed and are mainly in the form of outsourcing. Along with the progress of economic globalization and division of labor, they penetrate every aspect of the global market. On the other hand, Japan’s market remains relatively closed. Since World War II, the industrial structure of Japan has been continuously upgraded, and various industries, especially the services sectors, have entered into a relative mature stage and become the dominant industry. As for other countries, it is difficult to compete with Japanese companies because of the high trade barriers.

From the bottom two regions, the agriculture and mining sectors in underdeveloped countries/regions have lower generalist degree; except for achieving self-sufficiency (Region D), they only open international trade channels with the manufacturing sectors of a few developed economies (Region C). On the one hand, as the multilateral trading system is frequently challenged by unilateralism, agriculture sectors often passively become an important bargaining chip for balancing bilateral economic and trade relations, together with the presence of invisible barriers to agricultural trade, posing obstacles to the globalization process of the agricultural sector. On the other hand, since the globalization of the mining sectors depend on resource endowment and geographical factors, only a few countries are able to achieve significant exports of mineral resources, upon which most other countries have to rely.

### 5.2. Trend Analysis

To observe the dynamic trend, this paper put together the top twenty industrial sectors in terms of generalist degree on both sides, as shown in Figure 4. Overall, the major generalist sectors did not change significantly between 1990 and 2015, with the absolute generalist value fluctuating on a small scale. In particular, the manufacturing and services sectors of the U.S. and Germany in the export sectors, and their manufacturing sectors in the import sectors, have always maintained a high generalist degree, which means these sectors are deeply integrated into all parts of the global economic cycle. In addition, some variation trends also deserve more attention.

First, on the matter of the export trade reflected by the upstream sector’s generalist degree or the import trade reflected by the downstream sector’s generalist degree, these tend to wax and wane. It is clear that, with the scaling-up influence exerted by China’s manufacturing export trade, the generalist degree distribution of the upstream manufacturing sector has evolved from a “U.S.–Germany–Japan” tripolar pattern to a “China–U.S.–Germany” pattern [39]. Meanwhile, Chinese exports of trade in services have begun to narrow the gap with developed countries and have surpassed Japan.

Second, the rise of manufacturing sectors on the Chinese mainland and India have brought an impact on Taiwan. As one of the once “Four Asian Dragons”, Taiwan used to be a supply chain hub in Asia, except for Germany and Japan, for western countries. However, with the advent of the dividends of Chinese reform and their opening up, productive enterprises in Taiwan began to move to the Chinese mainland and overseas, leading to the significant decrease in influence of Taiwan’s manufacturing industry. Besides, in order to accelerate the development of the manufacturing industry, the Indian government has introduced a batch of relevant measures to stimulate investment and ease market access for foreign investment. Due to the blockade and restrictions imposed by the European and American markets on the Chinese market, a huge market such as India is taken as the preferred place for partial industrial transfer, which provides favorable conditions for the development of manufacturing industry in India [40].

Third, Serbia has become a new “European Factory” by virtue of its unique location and has started to play an important role in the import and export trade of manufacturing industry in recent years. Located at the junction of the East and the West, Serbia is an important hub connecting the major corridors of Europe and Asia and boasts strong connectivity. Besides, it has signed free trade agreements with the European Union and Central and Eastern Europe, and enjoys the most-favored-nation treatment from the U.S. With the progress of the Belt and Road Initiative, Chinese enterprise also brings infrastructure construction, creating a favorable environment for the development of the Serbian manufacturing industry. All of these positive factors make Serbia an important intermediate goods processing link in the GVC.

### 5.3. Stability Analysis

It is necessary to quantify the influence of generalist and specialist sectors on the nestedness of the GVC network, two control tests were designed to examine the influence of a certain sector and the cumulative influence of multiple sectors, respectively. The results are shown in Figure 5 and Figure 6.

After removing a generalist sector (see Figure 5), the NODF of the nested network significantly decreases, indicating that the higher the industry sector’s generalist degree, the more positive its effect on maintaining the stability of the GVC network. In contrast, after removing a specialist sector, the NODF slightly increases, which means that an industrial sector with a lower degree of generalist would weaken it. With the deepening of globalization, these industrial sectors risk being marginalized or even eliminated if they do not actively participate in international competition and cooperation.

Figure 6 further confirms the above findings. After removing a small proportion of industrial sectors with the highest generalist degree, the NODF of the nested network falls drastically, i.e., the stability of the GVC network deteriorates rapidly, indicating that a few generalist sectors are important hubs to maintain the functioning of the GVC. In contrast, when the industrial sectors with the lowest generalist degree are removed, NODF displays an increasing trend that does not start to decline until only 10% of the industry sectors are left. This reinforces the importance of the generalist sectors to stability.

By comparison of Figure 6a,b, we further find that there is a difference between generalist and specialist sectors in their terms of impact. Given the same proportion of removed generalist sectors (e.g., 10%), the removal of upstream sectors would exert a greater negative impact on the nestedness of the GVC network than that of downstream sectors. In other words, the global demand network of intermediate goods (consisting of downstream sectors) is sturdier than the supply network (consisting of upstream sectors). Considering this, when the global economy faces systematic risks, in order to cope with the resulting pressure or even disruptions of supply chains and reduced economic dependence on external resources, many countries have explored alternatives for supply chain management and import dependency. For example, they usually move their supply chains to countries less affected by the pandemic, pull some of the production capacity back from overseas or accelerate the industrialization process.

### 5.4. Evolutionary Mechanism

Dependency Theory, also known as Core-Periphery Theory, is established on the world trade pattern and the resulting unequal international division of labor. It is powerful as it explains the differences between developed and developing economies. The simple explanation is that developed countries gather at the center of the world economic system, while developing countries scatter at the periphery. Functionally, the countries at the center transfer the production of primary products to peripheral countries through capital import and transnational corporations, exploiting the peripheral countries’ cheap labor resources to develop labor-intensive industries and, thus, optimizing their own industrial structure. Being subject to the external constraints of the central countries, peripheral countries form dependence on the central countries and the surplus value keeps flowing from the periphery to the center, thus leading to the rich countries getting richer and the poor countries getting poorer [41,42,43,44]. For example, after World War II, the Asian, African and, especially, Latin American countries did not embark on the road to affluence after their attainment of independence but, instead, became even more dependent on, and formed the neo-colonialist industrial affiliation with, the economies of capitalist countries in Europe and North America. Given that, advocates of the dependency theory call for trade protection and import substitution in the peripheral countries and, with a strong nationalist tendency, encourage them to develop their own industries. However, this theory puts the peripheral countries in a passive position and attributes their economic distress on external factors, taking no consideration of the drawbacks in their domestic economic structures, which is to some extent pessimistic and biased.

Despite the similar nested structures of the mutual benefit ecosystem and the global economic system at the topological level, their formation mechanisms are not identical. Species in the mutual benefit ecosystem enhance their ecological benefits by continuously adjusting their interactions with other species, and the nested structure evolves through species’ dynamic game play and active adaptation to the environment. On the other hand, the driving force behind the formation of the core-periphery model of the global economic system lies in the international division of labor based on the countries’ comparative advantages and, therefore, features historical inevitability. However, from the perspective of dynamic development, peripheral countries are not always stuck in a position of being exploited and unable to develop their economies. On the contrary, the industrial transfer of the center countries creates opportunities for the peripheral countries to make full use of the capital and technology of developed countries to promote their own industrial development and technological innovation, thereby achieving so-called “Corner Overtaking”. With the increased depth and breadth of the GVC, by transferring many industrial production processes to developing countries, developed countries have completed the transition from a production-based society to a consumption-based society and need to rely on the supply of goods from developing countries. This has inevitably led to the emergence of industrial hollowing-out in some developed countries, increasing their dependency on developing countries as the world factories. Figure 7 briefly shows the movement of peripheral countries to the center of the industrial landscape, which is also the formation process of the nested structure of the global production network. From the perspective of evolutionarily economics, the continuously flattened world is derived by the evolutionarily stable equilibrium of the global production system.

Of course, the heterogeneity of economic development is still prevalent, and even increasingly serious. Even if the peripheral countries in the global industrial pattern achieve their economic growth targets, they are still at the end of high-tech diffusion, lacking core technologies and high value-added products, and are often subject to economic sanctions and technological blockade by central countries. In other words, the dependence of peripheral countries on the center countries is much stronger than that in the reverse. Developing countries therefore need to face up to the gap with developed countries in various aspects, transforming their economic growth mode so as to become the beneficiaries of economic globalization rather than just the contributors.

## 6. Econometric Analysis

Based on the above analysis, it is found that the generalist degree of a country’s industrial sectors is closely related to its economic status and productive capacity. Therefore, the focus of the following section is on whether a country’s economic condition is affected by the nested structure of the GVC network or, in other words, how a country’s macroeconomic performance is related to the microstructure of trade networks.

### 6.1. Correlation between Variables

Considering that a country’s macroeconomic performance is affected by both domestic and international trade cycles, this paper designs three NODF-based indicators to measure the nestedness of the local network in terms of economies. Firstly, DTN-NODF measures the nestedness of the home trade network, which consists of trade activities of intermediate products between industrial sectors within a country. Secondly, ETN-NODF measures the nestedness of the export trade network, which is formed when industrial sectors of a country, as upstream sectors (supply side), trades intermediate products with other countries. Thirdly, ITN-NODF measures the nestedness of the import trade network formed when a country’s industry, as a downstream sector (demand side), trades intermediate goods with other countries. In terms of macroeconomic performance, this paper uses Gross Domestic Product (GDP) data provided by the World Bank. The correlation diagrams of these are plotted at intervals of five years, as shown in Figure 8.

First, in all countries, the values of DTN-NODF are larger (mostly between 50 and 80), and those of ETN-NODF and ITN-NODF are smaller (mostly between 0 and 20). This is because, compared with international trade networks, most countries have relatively mature domestic trade networks, in which domestic industrial sectors can also form synergy. Hence, it is much less difficult and risky to form a domestic trade cycle of industrial chains, moving the original country-to-country trade to a province-to-province and city-to-city economic cycle. Certainly, the premise is that the country’s domestic market is sufficiently huge and the industrial system complete.

Second, no matter whether a domestic trade network or an international trade network, economies with better macroeconomic performance usually have higher nestedness. We believe that it is the relatively mature trade mechanisms that bring with them economic benefits and avoided risks. Hence, both domestic and international markets are equally important for a country’s economic development. How to better connect and utilize them is the key for countries to gain new advantages in international cooperation and competition.

Third, the ETN-NODF and ITN-NODF of the U. S. show a negative correlation with its GDP. In recent years, the U.S. has integrated a large amount of capital into the highly lucrative consumption side and shifted low-end manufacturing to countries with cheap labor costs, thus leading to the advent of manufacturing hollowing-out. Combined with industrial shocks from many developed (e.g., Germany and Japan) and developing countries (e.g., China), the international trade cycle does not seem to be contributing to the macroeconomic performance of the U.S. as expected. Such an industrial layout of the U.S. undermines its stability when encountering the rare but severe systematic risk. For instance, it is impressive that the White House could not obtain enough prevention and control supplies at the beginning of the COVID-19 pandemic.

### 6.2. Regression Model

To describe the quantitative relationship between GDP and the NODF-based indicators accurately, this paper applies regression analysis on these variables. First, a mixed effect regression model was established by taking the GDP of each country as the dependent variable and DTN-NODF, ETN-NODF and ITN-NODF as the independent variables, as shown in Table 1.

The correlation coefficients between the three independent variables were examined to check the existence of multicollinearity problems in the above model, so as to avoid spurious regression and ensure the validity of the model. The results show that the two variables, ETN-NODF and ITN-NODF, are significantly correlated, with correlation coefficients as high as 0.8608 (0.2631 for DTN-NODF and ETN-NODF, and 0.3289 for DTN-NODF and ITN-NODF).

To avoid multicollinearity problems in the following regression, Principal Component Analysis (PCA) was performed between ETN-NODF and ITN-NODF to investigate the components that constitute their covariance and ensure the orthogonality of the independent variables [45]. The ranked components with their loadings are listed in Table 2, revealing that both the KMO and the SMC confirm the correctness of PCA. The first principal components of PCA results are retained in this paper. Meanwhile, as shown in Appendix A, Table A5, we used a Ridge Regression to find the penalization term and found that the coefficients and conclusions are robust. We also provided the results of Fixed Effects model (FE), Random Effects model (RE) and Least Square Dummy Variable model (LSDV). As shown in Appendix A, Table A6, Table A7 and Table A8, we concluded that the pooled regression, i.e., Mixed Effect Regression model with PCA is the optimal solution (Appendix D).

As shown in Table 3, the model exhibits a relatively good fit with a large $R^{2}\;$ and all the independent variables (DTN-NODF and component variables) pass the *p*-value test at the significance level of 0.01.

From the above results, DTN-NODF displays a weak negative correlation with GDP. It is believed that an excessively nested domestic trade network may hinder a country’s economic development. Although a highly nested industrial layout can enhance the stability of the production system, it can also bring about problems such as lack of effective competition, path dependence in innovation and blocked channels for international cooperation, thereby hampering the country’s macroeconomic performance. On the other hand, ETN-NODF and ITN-NODF show a significant positive correlation with GDP. This can be attributed to the fact that for those who actively participates in the international trade cycle, they can complement each other by their own advantages and efficiently leverage resources in the international market, which leads to domestic socioeconomic development.

Besides, the regression model indicates that the positive effect of ETN-NODF on GDP is about two times greater than that of ITN-NODF. This is also a self-evident phenomenon. Larger ETN-NODF means a higher degree of nestedness in the export trade network structure, more orderly export mechanisms and more stable relevant channels, which together help to increase the trade surplus and boost domestic economic growth. At the same time, export growth drives import growth and secures the source of raw materials for the normal functioning of a country’s production system. As a result, this promotes the healthy development of the import trade network in turn, along with the increase of ITN-NODF. However, compared to the export trade network that can be unstoppably expanded, the expansion of the import trade network is limited by the relatively homogenous source of raw materials. That is why ETN-NODF is higher in value than ITN-NODF.

## 7. Conclusions

The vertical international division of labor and the continuous development of the global production network have played an unprecedented role in promoting the economy and trade of all countries in the world. The dynamic exchange of resources across the world is also an important guarantee for the stable and orderly progress of the GVC network. Through static and dynamic analysis of the generalist degree of the industrial sectors, the phenomenon is found that most of the generalist sectors come from more developed economies, which are in the most closely nested areas, and their internal competition is fierce. Due to the rapid increase in the generalist degree of China’s manufacturing industry, the manufacturing and services sectors in Japan and Taiwan have shown a downward trend, which has paved the way for changes in the global supply chain pattern. These advanced economies, as well as their industrial sectors, have also played a decisive role in maintaining the stability of the GVC network and promoting the process of global economic integration. In addition, there is still a lot of room for optimization of the industrial layout in the GVC. Encouraging specialist sectors to actively integrate into a global trade system is an effective way to realize this.

## Figures and Tables

**Figure 1 entropy-23-01077-f001:**
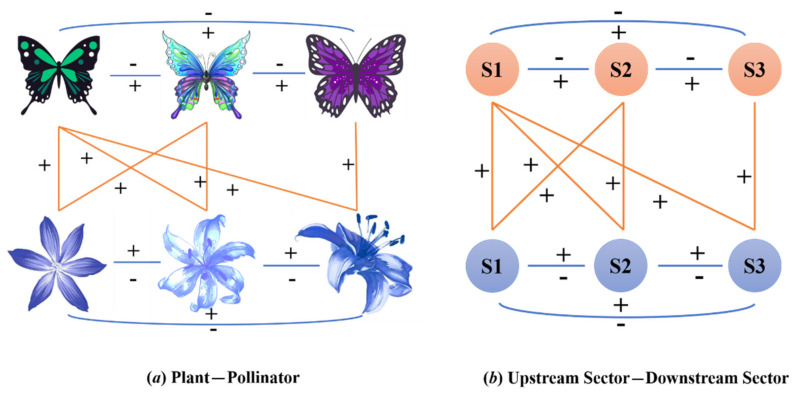
Comparison between Mutualistic System and Global Production System. (**a**) There exists a mutually beneficial symbiosis relationship between plants and pollinators. In simple terms, pollinators pollinate plants to promote the formation of their fruits and take in the nutrients they need at the same time. Among pollinators, there is not only competition for plants but also collaboration to complete the process of collecting pollination, which will be beneficial to both sides as their population grows. (**b**) Refers to the global production system where the orange circles represent the upstream sectors and the blue circles the downstream sectors. The numerous upstream and downstream sectors on the GVC cooperate to complete the industrial division of labor, while the upstream sectors not only compete for the same buyers but also collaborate to make sure these buyers can get what they need in the production process.

**Figure 2 entropy-23-01077-f002:**
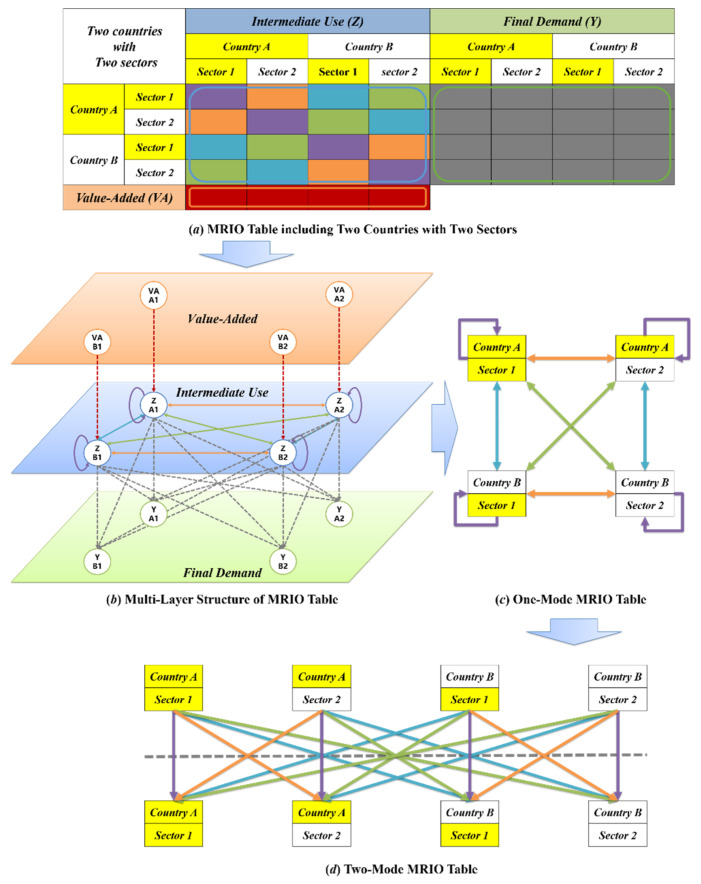
Relationship between MRIO Table and Network. Typical MRIO table includes three different areas, namely value-added, intermediate use and final demand. It is possible that the whole global economic system can be abstracted to a multilayer network as shown in Figure 2b, which includes three layers: the value-added layer, the intermediate use layer, and the final demand layer. The intermediate use layer can be further treated as a puzzle that is made of many single-layer networks out of a multilayer network, in which the nodes are the countries/regions, the layers are the industrial sectors, and links can be established from sellers to buyers within and across industrial sectors. If necessary, we can change the one-mode MRIO network into a two-mode network to separate the inner identity of each sector and prepare for the projection. In Figure 2d, the same sector distributes on the two sides of the dotted line, which means it belongs to both the upper stream and the lower stream. In other words, the upper stream sector in the MRIO table could be referred to as the object nodes in the bipartite graph, while the lower stream sector could be the participant nodes. Now, the self-loop is transformed into an edge between the two identities of this sector.

**Figure 3 entropy-23-01077-f003:**
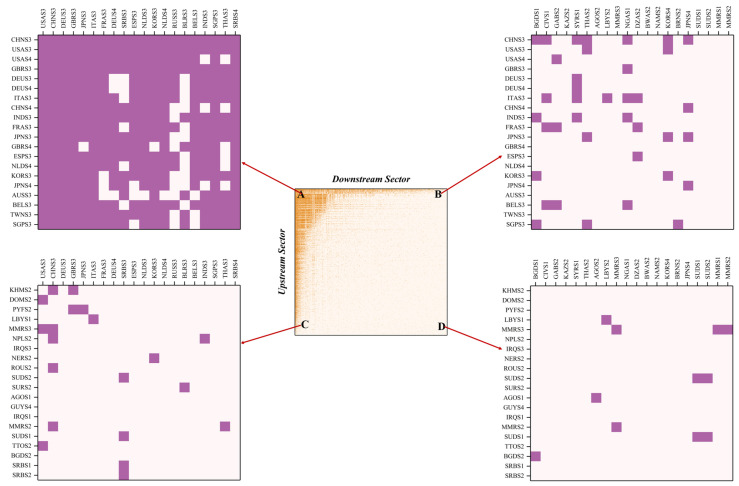
Topology of Different Regions after Sorting the Adjacency Matrix of GVC Network Based on SBD Algorithm. (**A**) The IO relationship between the upstream and downstream generalist sectors; (**B**) The IO relationship between the upstream generalist sectors and the downstream specialist sectors; (**C**) The IO relationship between the upstream specialist sectors and the downstream generalist sectors; (**D**) The IO relationship between the upstream and downstream specialist sectors.

**Figure 4 entropy-23-01077-f004:**
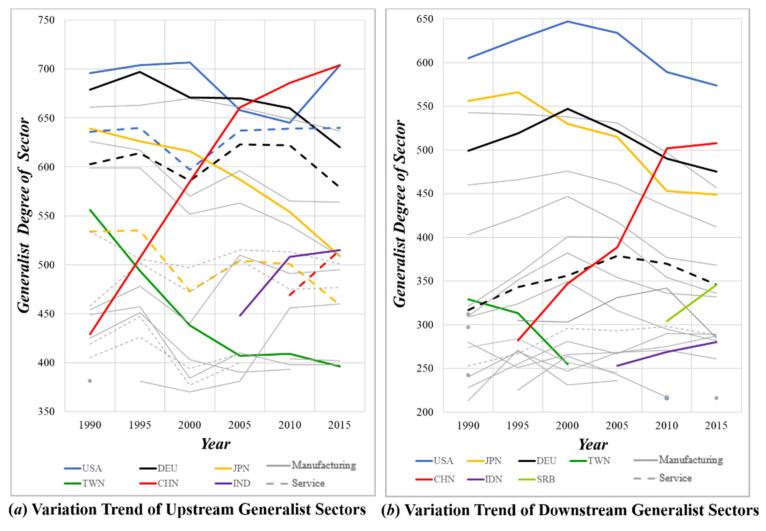
Generalist Degree Variation Trends of the Upstream and Downstream Industrial Sectors.

**Figure 5 entropy-23-01077-f005:**
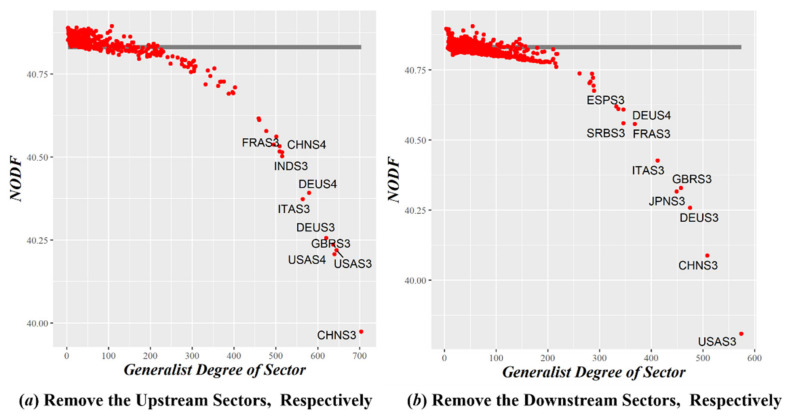
The NODF of Removing a Certain Industrial Sector of GVC Network. The horizontal gray lines represent the NODF of the nested network sorted by the SBD algorithm, and the red scatter points represent the correspondence between the generalist degree after removing a certain industrial sector (the size of the outdegree or indegree) and the new NODF of the nested network.

**Figure 6 entropy-23-01077-f006:**
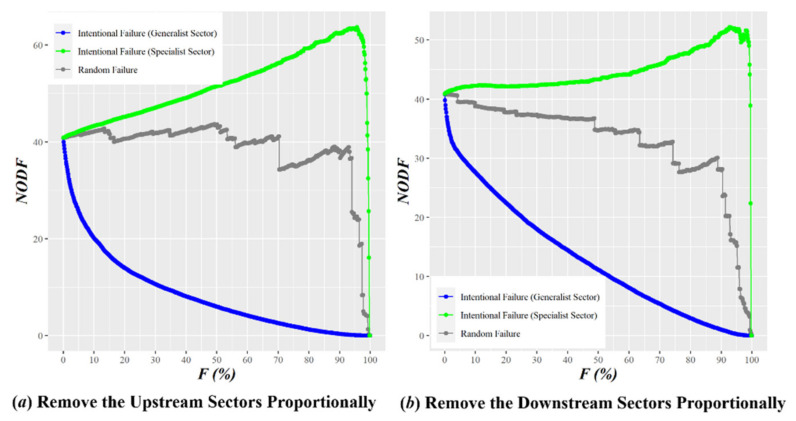
The NODF of Proportionally Removing Industry Sectors of GVC Network. The gray lines represent the variation in the value of network nestedness after randomly removing a certain proportion of industrial sectors from the aligned adjacency matrix; the blue lines represent the variation in the value of network nestedness after removing industrial sectors from the aligned adjacency matrix in the descending order of generalist degree; the green lines represent the variation in the value of network nestedness after removing industrial sectors from the aligned adjacency matrix in the descending order of specialist.

**Figure 7 entropy-23-01077-f007:**
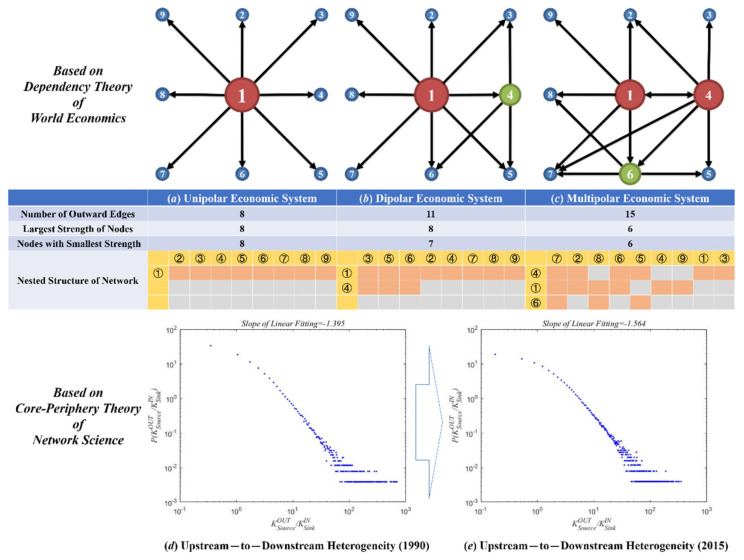
The Formation Process of Nested Structure of GVC Network. The orange circle 1 in (**a**) represents the developed countries initially at the center of the world economic system, while the yellow circles are the developing countries at the periphery, and the size of the circles reflects the degree of centrality of the countries; (**b**,**c**) indicate the gradual migration process of peripheral country 4 and peripheral country 6 to the central position, respectively. Besides, the ratio of outdegree of upstream sectors to indegree of downstream ones is designed to reflect the heterogeneity of development level of economies. Accordingly, the absolute value of slope of linear fitting increasing in (**d**,**e**) means our world is flattened by the economic integration.

**Figure 8 entropy-23-01077-f008:**
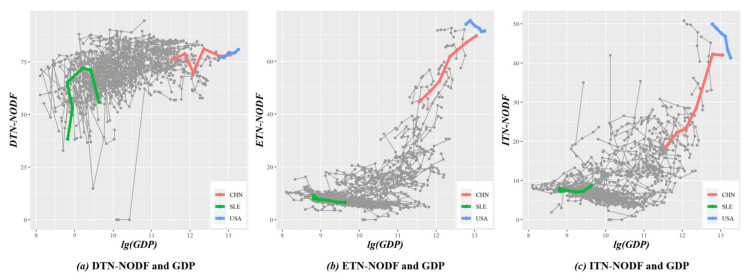
Correlation of DTN-NODF, ETN-NODF, ITN-NODF and GDP. Three countries with large differences in GDP are selected, with red representing China, blue the United States and green Sierra Leone. Data source: World Bank—https://data.worldbank.org.cn/indicator, accessed on 14 January 2021.

**Table 1 entropy-23-01077-t001:** Results of the Mixed Effect Regression Model.

Variables	Coef.	Robust Std. Err	t	*p*	95% Confidence Interval
DTN-NODF	−9.49	2.64	−3.60	0.000	[−14.67, −4.32]
ETN-NODF	44.27	3.88	11.40	0.000	[36.65, 51.90]
ITN-NODF	21.03	6.88	3.06	0.002	[7.54, 34.52]
Intercept Term	71.97	180.36	0.40	0.690	[−281.95, 425.89]
$R^{2}$ (adjusted)	0.411		Root MSE	867.68	

**Table 2 entropy-23-01077-t002:** Principal Component Analysis of ETN-NODF and ITN-NODF.

Component	Eigenvalue	Difference	Proportion	Cumulative	KMO	SMC
ETN-NODF	233.326	220.745	0.9488	0.9488	0.9999	0.7410
ITN-NODF	12.581	-	0.0512	1.0000	0.9999	0.7410

**Table 3 entropy-23-01077-t003:** Results of Mixed Effect Regression Model after PCA.

Variables	Coef.	Robust Std. Err	t	*p*	95% Confidence Interval
DTN-NODF	−9.63	2.60	−3.71	0.000	[−14.72, −4.54]
Comp.	48.99	1.83	26.74	0.000	[45.39, 52.58]
Intercept Term	74.09	180.14	0.41	0.681	[−279.40, 427.57]
$R^{2}$ (adjusted)	0.411		Root MSE	867.3	

Notes: Comp. = 0.8824ETN-NODF + 0.4705ITN-NODF.

## Data Availability

We signed a confidentiality agreement with the transportation company who provided us with the data used in this work. Hence the data will not be shared.

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
