# Peer review of "Nestedness-Based Measurement of Evolutionarily Stable Equilibrium of Global Production System"

_entropy, 2021, doi:10.3390/e23081077_

Round 1
Reviewer 1 Report
Strongly inspired by the background of economical science, the authors deliberately explored huge data sets covering various industrial sectors of intra- and inter-national viewpoints. Their focal scope is nested structure commonly observed in biological systems with term of food-chain.
Although I couldn’t find any substantially new novelty from the rigorism of science and applied mathematics, what they reporting might be somehow meaningful from practical standpoint and application. So thus, I have rather a positive impression on the MS.
Yet, I would give several suggestions as below, which should be appropriately responded in the revised MS.
#1.
Their concept of ‘nestedness’ is fine. But it comes back to the nested structure in biological eco-system, a nested structure is multi-layer, that is plural not singular. I read that what the authors defined with the term of nestedness recognizes a simple two-body relation in a complex network, visually grasped as in Figs. 3 to 5. What they presumed seems to me ‘pair-wised’ relation that would not be same as ‘nested structure’. I do need, and the audience perhaps does as well, more clear explanation and justification on the point abovementioned.
#2.
Although the title suggests ‘equilibrium’ of an evolutionary process, there is less content on the point, especially there is almost none of any mathematical, i.e., quantitative, discussion. I understand that, from the authors’ standpoint, somehow qualitative one might be enough. Yet, some general review on an evolutionary dynamical system’ should be added to Introduction part by citing some recent books, such as; (i) Sociophysics Approach to Epidemics, Springer, 2021, (ii) Evolutionary Games with Sociophysics: Analysis of Traffic Flow and Epidemics, Springer, 2019, (iii) Fundamentals of Evolutionary Game Theory and its Applications, Springer, 2015.
Reviewer 2 Report
In general I have enjoyed reading this rather interesting manuscript. Looking through the ecological prism into economical networks provides an interesting perspective.
Yet I have found the paper extremely hard to read: some sentences are overly long (they could be split into multiple sentences to improve readability), others are structured in a confusing manner (e.g., the first sentence in the introduction).
I would also think that some of the parts are unnecessary and perhaps trivial. Especially comparison of the sorting algorithms. It seems to be a purely technical question without any actual relevance to the main result of the paper.
Literature review mostly goes through what was done previously by other groups. This is fine, but the problem I have is that the review fails to explain the relevance and/or intuition that the reviewed works bring.
It would be nice to see a clear explanation of the intuition behind the nestedness concept. What does it mean for the global economic system to be nested? Why it should be nested? Why is it important for a system to be nested?
The manuscript seems to use term "generalist degree", which doesn't seem to be well explained. How is it different from an ordinary degree? Why there is no mention of "specialist degree"?
Round 2
Reviewer 1 Report
The revised MS seems adequate for publication now...
Reviewer 2 Report
I feel that the authors have properly addressed issues I have raised. I see no further issues and therefore recommend to accept the revised manuscript for publication.